# COCO-LC: Colorfulness Controllable Language-based Colorization

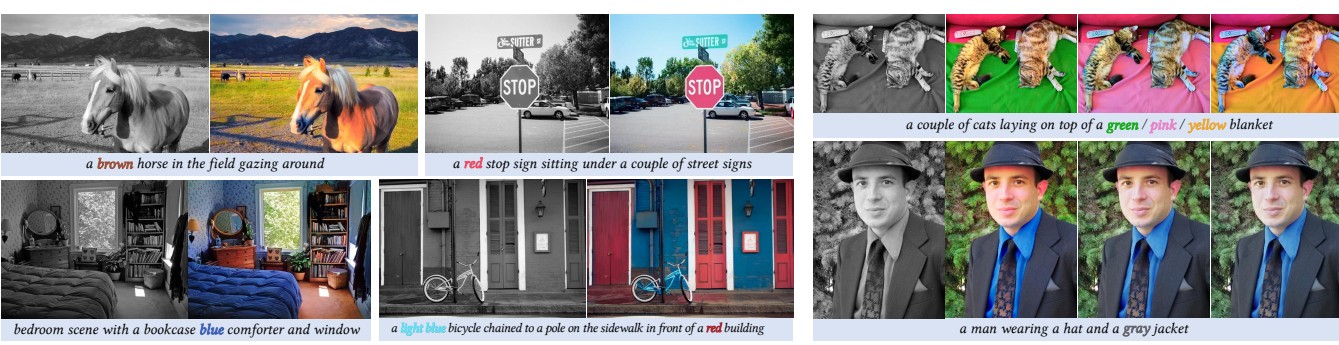

(a) our language-based image colorization results

(b) controllable colorization

**Figure 1: We propose COCO-LC, a novel colorfulness controllable language-based colorization framework. (a) COCO-LC generates realistic and semantic-consistent colorization results. (b) COCO-LC allows for flexible user control over (top) color types and (bottom) color styles.**

## ABSTRACT

Language-based image colorization aims to convert grayscale images to plausible and visually pleasing color images with language guidance, enjoying wide applications in historical photo restoration and the film industry. Existing methods mainly leverage large language models and diffusion models to incorporate language guidance into the colorization process. However, it is still a great challenge to build accurate correspondence between the gray image and the semantic instructions, leading to mismatched, overflowing and under-saturated colors. In this paper, we introduce a novel coarse-to-fine framework, COlorfulness COntrollable Language-based Colorization (COCO-LC), that effectively reinforces the image-text correspondence with coarsely colorized results. In addition, a multi-level condition that leverages both low-level and high-level cues of the gray image is introduced to realize accurate semantic-aware colorization without overflowing colors. Furthermore, we condition COCO-LC with a scale factor to determine the colorfulness of the output, flexibly meeting the different needs of users. We validate the superiority of COCO-LC over state-of-the-art image colorization methods in accurate, realistic and controllable colorization through extensive experiments.

## CCS CONCEPTS

• **Computing methodologies** → **Reconstruction**; **Image processing**; **Computational photography**.

## KEYWORDS

Colorfulness control, language guidance, image colorization

## 1 INTRODUCTION

Color plays a pivotal role in shaping human perception of the world. It serves not only as one of the most expressive visual elements, but also directly influences people's emotions through various color styles. Image colorization aims to convert grayscale images into color images, which has a wide applications in diverse fields such as old photo restoration, color grading, and automatic animation colorization. However, image colorization is an ill-posed problem as it involves inferring a three-channel color image from a single-channel illuminance image, which may have multiple reasonable solutions. Despite many automatic colorization methods [9, 19, 23, 25, 35, 47] have been proposed, most of them suffer such color ambiguity, leading to under-saturared colors with the mean of all possible color choices. To address this issue, conditional colorization begins to attract researchers' interests. By imposing additional constraints, such as language [5–7, 16, 41, 45], scribble [16, 44], reference image [27] and palette [38, 42], the method can more accurately render specified colors.

Recent advancement on large language models (LLMs) and diffusion models empowers image processing with language guidance. Compared with other visual conditions, text descriptions are highly informative, and are simple and efficient to use. Recent language-based image colorization methods [5, 7, 16, 41] either train language-vision aligned models supervised by dense human annotations, which may overfit on a small dataset and result in

poor quality; or employ pretrained large models such as BERT [11] and CLIP [31] to align language domain and image domain, which, however, is less effective in building an accurate correspondence between gray instances and text description.

Based on the above analysis, we summarize three key requirements lies on the image colorization problem: **(1) Realism.** The colorization results should look realistic. **(2) Consistency.** The colorization results should match the semantic content of the original gray images. **(3) Controllability.** Users can flexibly adjust the color of the output. Existing methods can only fulfill one or two requirements, hardly satisfying all above standards: The unconditional ones provide no controllability and suffer under-saturation, while the conditional ones are less flexible or fail to render semantically correct colors, leading to color overflow and inconsistency.

To build a powerful image colorization framework that simultaneously achieves realism, consistency and controllability, we propose **COCO-LC** for **CO**lorfulness **CO**ntrollable **L**anguage-based **C**olorization. To strengthen realism, our key idea is a coarse-to-fine framework that leverages Stable Diffusion [33], the pre-trained cross-modality generative model, to utilize its high capability for textual-visual modeling and powerful generative prior on the natural image distribution. To maintain the semantic consistency between gray inputs and colorization results, we further incorporate robust multi-level conditions with both low-level and high-level cues of the gray inputs. Finally, in order to provide more controllability, besides color types, we develop a novel colorfulness-controllable colorization decoder to produce results with diverse color styles ranging from vintage, realistic, to fantastic.

Specifically, we propose a coarse-to-fine training framework. In the coarse colorization stage, we inject features from the large-scale cross-modality model CLIP [31] into the latent space of the gray image. On this basis, the resulting latent codes with rich color information are used to guide the diffusion model to generate fine-level colorization results, which achieves accurate correspondence between the textual color words and visual gray instances.

Additionally, we propose multi-level conditions that effectively alleviate color overflow and mismatch issues. We design a novel dual-branch feature extractor that aligns the feature granularity of the low-level edge features with diffusion features for balanced condition injection. A semantic-aware feature regularization is further proposed to provide high-level features to improve semantical correspondence.

In terms of controllability, brightness, hue and saturation are three key characteristics to determine color values. While brightness is determined by the gray image and the control of hue is well studied for conditional colorization, controlling saturation remains less explored. To this end, we design a colorfulness-controllable colorization decoder, with a scaling factor to allow users to flexibly choose different color styles from vintage to gorgeous styles.

With the novel designs above, COCO-LC comprehensively realizes realism, consistency, and controllability. Extensive experimental results demonstrate our superiority in generating high-quality color images over both automatic and language-based state-of-the-art baselines. In summary, our contributions are threefold:

- We propose a novel coarse-to-fine COCO-LC framework for language-based colorization that achieves high realism,

consistency, and controllability simultaneously. We leverage cross-modality model CLIP to build accurate correspondence between the color words and gray instances, significantly improving the visual-textual color consistency.

- We develop multi-level conditions with both low-level and high-level guidance to find accurate color-semantic correspondence. A novel dual-branch feature extractor is proposed to align the feature granularity for balanced condition injection, which effectively alleviates color overflow.

- We design a colorfulness-controllable decoder, which adaptively fuses the predicted color information and the original grayscale information, allowing users to choose fantastic, realistic or vintage colorized results to flexibly satisfy diverse user preferences.

## 2 RELATED WORKS

### 2.1 Automatic Colorization

Automatic colorization aims to generate plausible colorful images from grayscale input without user guidance. Cheng *et al.* [9] introduced the first deep-based colorization model. In view of the imbalance of the color space, CIC [47] defines a classification loss in the quantized *ab* color space rather than the traditional regression loss, leading to more saturated results. Researchers produce models with more plausible colorization results with Pixcolor [12]. With the development of generative models, many researchers began to make efforts on colorization problem with GAN [10]. Cao *et al.* [4] conditioned GAN with grayscale information in multiple layers to maintain spatial feature consistency. ChromaGAN [37] adds color error and class distribution losses to optimize training for colorization. BigColor [23] improves vividness through pre-trained generative priors while suffering from color cast. DeOldify [1] utilizes U-Net as the generator model with a self-attention scheme and reduces direct GAN training time with a scheduled adversarial loss. Furthermore, to utilize the long range dependency and increase model capacity, transformer-based architectures [36] are widely used in colorization tasks [19, 20, 25]. ColorFormer [19] designs a color memory module to learn and store the semantic-color mapping. HistoryNet [20] introduces fine-grained semantic understanding and classification prior to achieve accurate colorization and prevent color overflow. Priors including class labels [18], instance bounding boxes [35], and semantic segmentation maps [49] are introduced to further guide colorization models with semantic information. However, unconditional colorization is an ill-posed problem, resulting in color ambiguity and under-saturation. From the users' perspective, lack of controllability limits the practicability of this kind of approaches.

### 2.2 Language-based Colorization

With the help of the flexibility of text descriptions, language-based colorization methods enable simple but effective control over instance colors. Unicolor [16] developed a unified framework to support colorization in multiple modalities, including text descriptions. It proposed a spatial partitioning heuristic method to align color words and instances by leveraging CLIP [31] as zero-shot classifier, but struggle with finer colorization ability in spatial, and can only

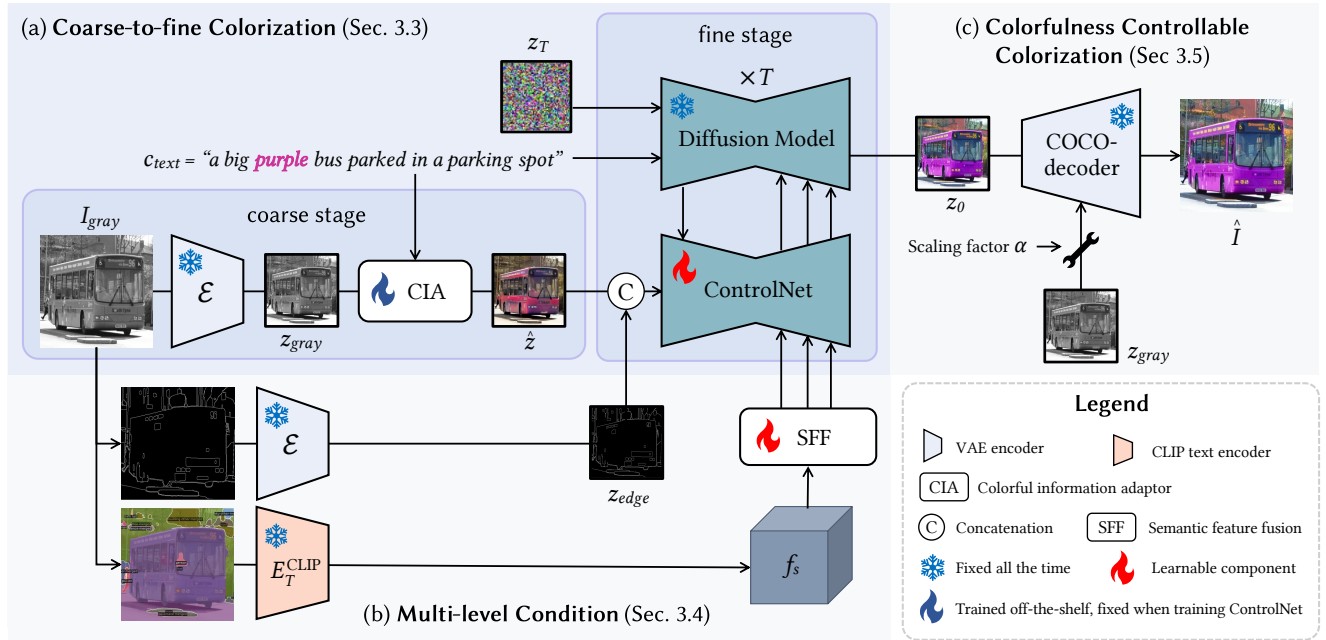

**Figure 2: Illustration of the proposed COCO-LC framework with three key components: (a) coarse-to-fine colorization; (b) consistency-aware multi-level condition; (c) colorfulness controllable colorization.**

deal with several classic colors. L-CoDe [41] introduced an color-object correspondence matrix and an attention transfer module to address the color-instance coupling problem, along with a soft-gated injection module for resolving the color-object mismatch issue. L-CoDer [5] introduced transformers [36] into language-based colorization to deal with inaccuracy while keeping the language decoupling property. However, both L-CoDe [41] and L-CoDer [5] need to create an additional color-instance decoupling module to align colors with gray instances. This module have to train on extra human annotations, resulting in more training cost. Thus, L-CoIns [7] achieved instance awareness with the grouping mechanism to adaptively aggregate similar image patches without additional annotations. Furthermore, L-CAD [6] reduces the dependence of colorization on the precision of language guidance by using a pretrained diffusion model with rich color priors and superior cross-modal capability. However, it lacks the ability to generate high-resolution colorized images due to the cost of fine-tuning.

### 2.3 Diffusion-based Image Colorization

Recently, diffusion model shows its advantages in image generation [32, 33], image editing [34] and image restoration [29, 43]. As diffusion models hold high-quality color and semantic priors with the benifit from large-scale pretraining, L-CAD [6] designs a luminance-guided compression module and merge latent codes of grayscale images into diffusion process to maintain accurate structure when fine-tuning the Stable Diffusion [33]. Diffusing Colors [45] adopts the cold diffusion mechanism [2] to learn a colorization process iteratively. Besides, CtrlColor [28] develops multiple encoders based on ControlNet [46] according to different kind of conditions, such as user scribbles, reference images, regions or text descriptions. As colorized results need to spatially align with

input grayscale images, diffusion-based colorization methods have to balance the trade-off between realism and consistency. Overall, diffusion-based image colorization is still a cutting-edge and challenging topic.

## 3 COCO-LC

### 3.1 Preliminary: Stable Diffusion

Diffusion models learn image distributions based on a diffusion process and a denoising process, where Stable Diffusion [33] operates in the latent space with a VAE encoder $\mathcal{E}$ and a VAE decoder $\mathcal{D}$. During the diffusion process, random Gaussian noises $\epsilon \sim \mathcal{N}(0, I)$ are gradually added to the encoded latent feature $z_0 = \mathcal{E}(x_0)$ of the input image $x_0$ in $T$ steps,

$$z_t = \sqrt{\alpha_t} z_0 + \sqrt{1 - \alpha_t} \epsilon, \qquad (1)$$

producing a series of noisy samples $z_1, ..., z_T$. As $z_T$ can be treated as a standard Gaussian noise approximately when $T$ is large enough, the denoising process can recover a realistic image $x_0 = \mathcal{D}(z_0)$ from a srandard Gaussian noise iteratively, which is achieved by training a neural network U-Net $\epsilon_\theta$ with parameter $\theta$ to predict $\epsilon$ at each timestep $t$ based on $z_t$ with the loss function:

$$\mathcal{L} = \mathbb{E}_{\epsilon \sim \mathcal{N}(0,I), t}[\|\epsilon - \epsilon_\theta(z_t, c_{text}, t)\|^2], \qquad (2)$$

where in Stable Diffusion, text conditions $c_{text}$ are taken as guidance to constrain the generation through cross-attention mechanism. Besides text, Stable Diffusion can be addtionally conditioned on images with ControlNet [46]. ControlNet is a trainable copy of the diffusion model, serving as a side branch to accept and apply image conditions $c_i$ to the main diffusion branch. The overall learning objective can be formulated as

$$\mathcal{L} = \mathbb{E}_{\epsilon \sim \mathcal{N}(0,I), t}[\|\epsilon - \epsilon_\theta(z_t, c_{text}, c_i, t)\|^2]. \qquad (3)$$

It is natural to apply ControlNet to the colorization task, *i.e.*, using grayscale image $I_{gray}$ as $c_i$ to predict the corresponding colorful image $I$. However, we experimentally found that vanilla ControlNet is not competent to offer high-quality gray image constraints. ControlNet uses simple convolution layers to preprocess the conditional images, leading to imbalanced feature granularity between the ControlNet and diffusion branches and causing structure distortion. It also fails to build robust correspondence between the gray instances and the color words, resulting in color mismatch and color overflow. This paper propose a new croase-to-fine framework with novel multi-level consistency-aware conditions to solve the above problems.

## 3.2 Overview architecture

COCO-LC takes as input a grayscale image $I_{gray}$, a text prompt describing the desired color of the instances in $c_{text}$, and a scaling factor $\alpha$ to indicate the targeted colorfulness, and produce a corresponding colorization result $\hat{I}$. As shown in Fig. 2, the proposed COCO-LC framework consist of three key components: (1) coarse-to-fine colorization; (2) consistency-aware multi-level condition; (3) colorfulness controllable colorization decoder.

Our coarse-to-fine colorization (Sec. 3.3) first colorizes the latent feature $z_{gray} = \mathcal{E}(I_{gray})$ with our proposed lightweight Colorful Information Adaptor (CIA). CIA leverages the power of CLIP [31] to create a coarsely colorized feature based on $c_{text}$ and $z_{gray}$. The resulting $\hat{z}$ is fed into the main diffusion branch by our condition injection branch for fine-level colorization.

For consistency-aware multi-level condition, we extract the edge map and segmentation map from $I_{gray}$ to serve as low-level structure and high-level semantic guidance, respectively. We merge the low-level edge maps and high-level semantic segmentation maps with the condition injection branch, as will be detailed in Sec. 3.4.

In Sec. 3.5, we will introduce our colorfulness controllable colorization decoder (COCO-decoder), that adapts the VAE decoder $\mathcal{D}$ to merge grayscale image and provides a scaling factor $\alpha$ to enable users to flexibly adjust the colorfulness of the output.

## 3.3 Coarse-to-fine Colorization Framework

While Stable Diffusion demonstrates satisfying performance of matching color words and image instances with cross-attention mechanism, it is not trival to find proper correspondence when the instances lie in the gray feature space because of the domain gap between the grayscale images and the color images. Our coarse-to-fine colorization framework solve this issue by gradually narrow the domain gap in two stages.

*3.3.1 Coarse colorization.* We design a Colorful Information Adaptor (CIA) to merge color information into VAE latent space, as shown in Fig. 3. This lightweight adptor is trained off-the-shelf and kept fixed during fine colorization stage, enables its flexibility and simplicity. During training, given a color image $I$, we obtain its grayscale version $I_{gray}$ and map images into feature domains with VAE encoder $\mathcal{E}$ and CLIP image encoder $E_I^{CLIP}$. CIA takes the gray image latent feature $\mathcal{E}(I_{gray})$ and CLIP color image feature $f_I = E_I^{CLIP}(I)$ as input, merging the grayscale content information and color style information by a scale-shift operation, which can

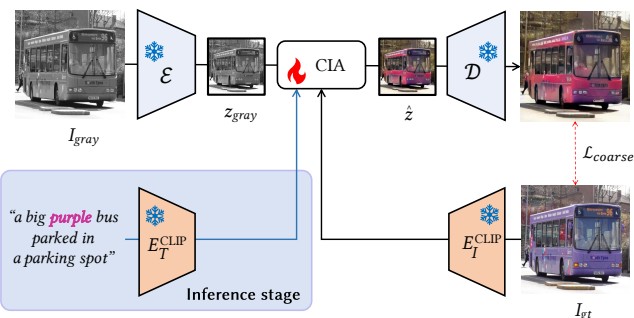

**Figure 3: Illustration of Colorful Information Adaptor (CIA). We utilize six ResBlocks to merge grayscale structure information and color style information hierarchically, following a paramerized version of AdaIN [15].**

be formulated by Eq. (4):

$$\hat{z} = \frac{z - \mu(z)}{\sigma(z)}\left(1 + F_{scale}(f_I)\right) + F_{shift}(f_I), \quad (4)$$

where $F_{scale}, F_{shift}$ denote the mapping networks in CIA to wrap the color information into scale-shift parameters. $z$ is the structure feature initialized with $\mathcal{E}(I_{gray})$. $\mu(z)$ and $\sigma(z)$ denote the mean and standard deviation of $z$. This fusion operation can be also understood as a parameterized version of AdaIN [15], and is applied to $z$ six times. The training objective of CIA is

$$\begin{aligned}\mathcal{L}_{coarse} = &\left\|\mathcal{D}(CIA(\mathcal{E}(I_{gray}), f_I)) - I\right\|^2 \\ &+ \lambda \cdot VGG\_Loss(\mathcal{D}(CIA(\mathcal{E}(I_{gray}), f_I)), I),\end{aligned} \quad (5)$$

where VGG_Loss is the perceptual loss [21] and we simply set $\lambda = 1$.

During testing, as the color image $I$ is unavailbale, we utilize CLIP text encoder $E_T^{CLIP}$ and replace $f_I$ with the encoded text description $E_T^{CLIP}(c_{text})$, thanks to the alignment of the CLIP visual and textual spaces. Moreover, excluding other information that is not related to color in image features (such as texture, luminance), $c_{text}$ with color cues has greater color information density, further enhance the reasonability of the coarse colorization results. Thus, we have $\hat{z} = CIA(\mathcal{E}(I_{gray}), E_T^{CLIP}(c_{text}))$. We also try to use $E_T^{CLIP}(c_{text})$ during training CIA, but find it become hard to converge, likely due to the high sparsity of the language domain.

*3.3.2 Fine colorization.* CLIP features cannot represent local informations of image in a finer granularity, thus the coarse colorization results only provide rough color-instance correspondence. Then in the fine stage, on the basis of the semi-colorized latent $\hat{z}$, we use the diffusion model to match the color words and instances precisely. The fine-stage objective function is

$$\mathcal{L}_{fine} = \mathbb{E}_{\epsilon \sim \mathcal{N}(0,I),t}\left[\left\|\epsilon - \epsilon_\theta(z_t, c_{text}, \hat{z}, z_{edge}, f_s, t)\right\|^2\right], \quad (6)$$

where $z_{edge}$ and $f_s$ denotes edge and semantic map feature conditions extracted from $I_{gray}$, which will be introduced in Sec. 3.4.

## 3.4 Consistency-aware Multi-level Condition

*3.4.1 Low-level dual-branch condition insertion.* As has been analyzed in Sec. 3.1, vanilla ControlNet has imbalanced feature granularity with the diffusion branch, which is especially harmful when

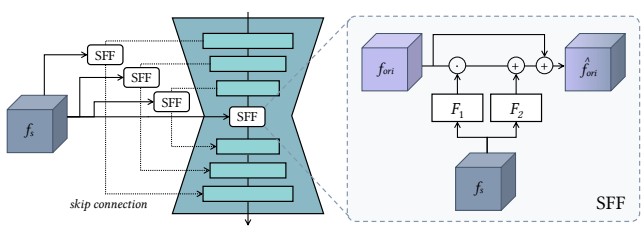

**Figure 4: Illustration of high-level feature regularization**

dealing with gray image conditions, since grayscale images hold rich structure and semantic information than simple conditions like depth maps, and Canny edge maps [46]. Vanilla ControlNet do not have the ability to extract enough semantic features to precisely control the diffusion generation, leading to color overflow, color mismatch and structure distortion.

To balance the feature granularity between condition injection and diffusion generation, we propose a dual-branch feature extractor based on $\mathcal{E}$. Given a grayscale image $I_{gray}$, we utilize SAM [24] as a zero-shot edge detector to extract its instance-aware edge map $I_{edge}$, and encode it into the latent space $z_{edge} = \mathcal{E}(I_{edge})$. As mentioned before, we have inserted color information into $z_{gray}$ with CIA, to obtain $\hat{z}$ with rich color priors. We concatenate $\hat{z}$ and $z_{edge}$, and fuse them using a single convolution layer. We use $\tilde{z} = Conv([\hat{z}, z_{edge}])$ as the input of the ControlNet branch.

*3.4.2 High-level semantic feature modulation.* To provide high-level feature regularization to build more accurate correspondence between the color and the instances, we leverage Mask2Former [8] to predict semantic segmentation maps based on $I_{gray}$. As the semantic map assigns labels to each pixel, we treat each label as a text description and use CLIP text encoder $E_T^{CLIP}$ to extract the feature of each description appear in semantic maps. Finally, we aggregate these text features into a standard spatial semantic feature $f_s \in \mathcal{R}^{c \times h \times w}$, where $c$ represents the dimension of CLIP text features and $h \times w$ is the resolution of $I_{gray}$.

We design a semantic feature fusion block (SFF) to modulate the original features $f_{ori}$ on the skip connection of U-Net with $f_s$, as illustrated in Fig. 4. SFF adapts SPADE [30] with minor changes, and uses two submodules $F_1$ and $F_2$ to process and fuse the features in a "scale-shift" manner. This fusion operation can be formulated as

$$\hat{f}_{ori} = f_{ori} + LN(f_{ori}) \cdot F_1(f_s) + F_2(f_s), \tag{7}$$

where $LN$ is the Layer Normalization. $F_1$ and $F_2$ denote two mapping network to wrap $f_s$ into scale-shift parameters, with some convolution layers and downsampling layers.

## 3.5 Colorfulness Controllable Colorization

To generate colorization results with varying color richness according with the user preference, we present colorfulness controllable colorization decoder $\mathcal{D}_{COCO}$ (COCO-decoder), based on the VAE decoder $\mathcal{D}$ with a scaling factor $\alpha$. As shown in Fig. 5, $\mathcal{D}_{COCO}$ maintains a fixed decoder $\mathcal{D}$, a trainable decoder $\hat{\mathcal{D}}$ and trainable zero-initialized convolution layers $F_0$. We feed the gray image latent feature $z_{gray}$ into $\mathcal{D}$ and get a set of middle features $\{d_i\}$ as the structure guidance, where $i$ is the layer index. Correspondingly, we feed the diffusion output $z_0$ in the fine colorization stage into $\hat{\mathcal{D}}$,

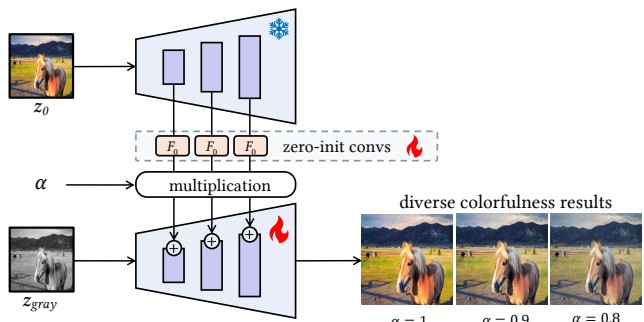

**Figure 5: The struct of Colorfulness Controllable (COCO) module which merge the feature space of two VAE decoder space. We extract the middle features during the decoding of gray latents, wrap them with zero-initialized convolution layers and add them with the middle features of the decoding of colorized latents.**

and obtain the corresponding features $\{\hat{d}_i\}$. In $\mathcal{D}_{COCO}$, we update $\hat{d}_i$ with $d_i$ by

$$\hat{d}_i \leftarrow \hat{d}_i + \alpha F_0(d_i). \tag{8}$$

During the training of $\mathcal{D}_{COCO}$, we set $\alpha = 1$ and optimize

$$\mathcal{L}_{COCO} = \mathbb{E}_I[\|\mathcal{D}_{COCO}(z_0, z_{gray}, \alpha) - I\|^2]. \tag{9}$$

Note that we use zero-initialized convolution layers to warm up the middle features during injection, preserving the capability of the pre-trained VAE decoder.

In the inference phase, we can use $\alpha$ to control the injection of gray information, *i.e.*, we can get diverse colorfulness results ranging from fantastic and gorgeous (low $\alpha$), bright and realistic (middle $\alpha$) to grayish and vintage (high $\alpha$), enabling users to choose the best colorized result according to their preference.

## 4 EXPERIMENTS

### 4.1 Implementation Details

*4.1.1 Training.* We train CIA with a single NVIDIA RTX 2080Ti GPU for 280k iterations with a batch size of 24. After that, we train COCO-LC on a single NVIDIA RTX 4090 GPU for 150k iterations with a batch size of 16. We use AdamW optimizer with $\beta_1 = 0.9$ and $\beta_2 = 0.999$. The learning rate is set to $10^{-5}$.

*4.1.2 Inference.* We use a single NVIDIA RTX 2080Ti GPU for inference. All testing images are resized to $512 \times 512$ with bilinear interpolation. Besides, we transform the output image to $LAB$ space and replace its $L$ channel with that of $I_{gray}$, to maintain the structure consistency. Our method will generate colorized results of the input grayscale images based on users' text description. If users don't provide any text descriptions, we utilize BLIP [26] to get a standard text description and do colorization automatically.

### 4.2 Evaluation

*4.2.1 Training data.* We conduct our experiments on language-based colorization datasets proposed by L-CoDe [41] and L-CoIns [7]: (i) the extended COCO-Stuff dataset, which is built upon the COCO-Stuff dataset [3] by discarding unqualified samples for the colorization task. We further filter out some black and white photos, remains

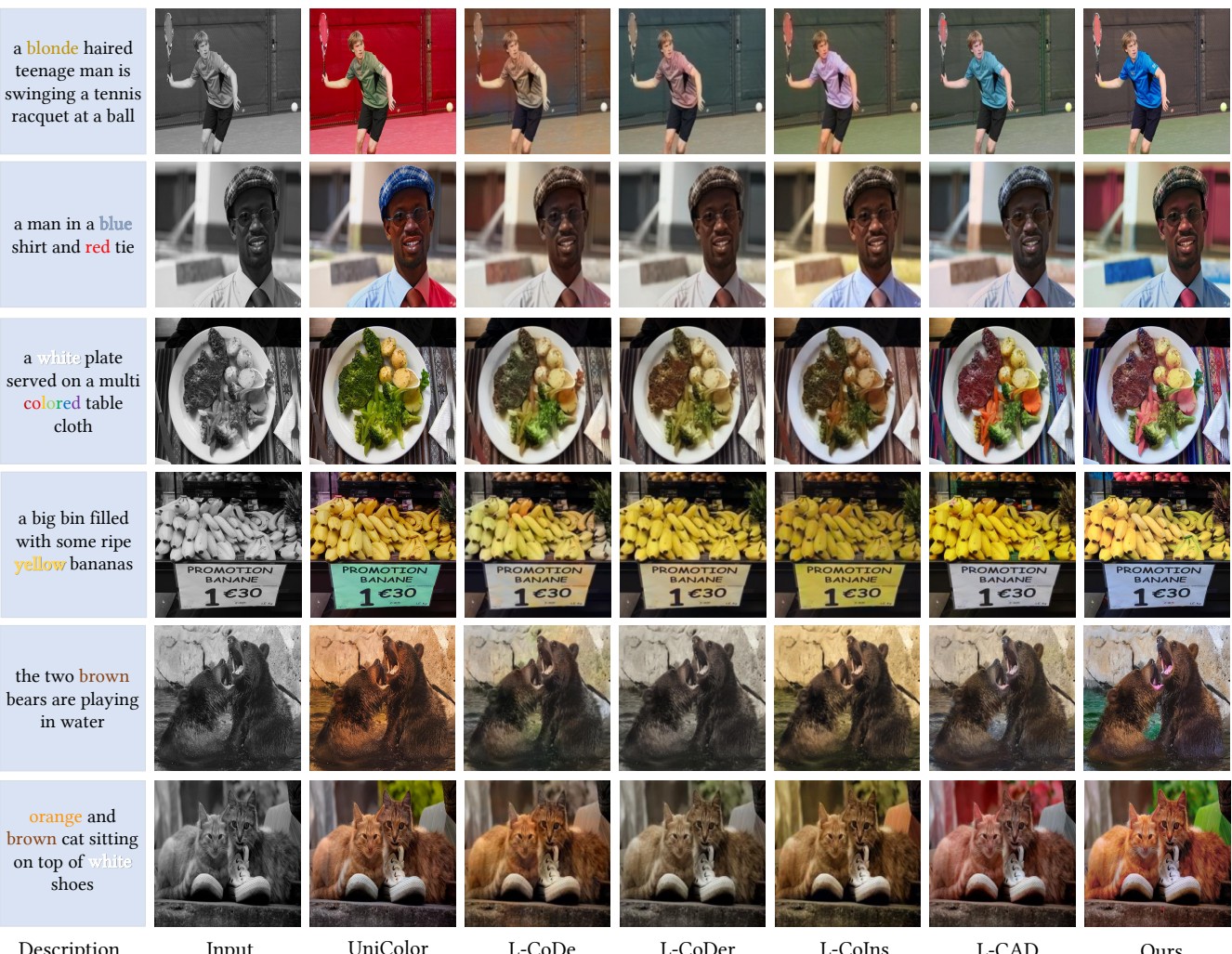

| Description | Input | UniColor | L-CoDe | L-CoDer | L-CoIns | L-CAD | Ours |

**Figure 6: Qualitative comparison results of our methods and other language-based methods. Our colorized images enjoy a higher saturation and more plausible visual effect. Zoom in for better visualization.**

56K training images and 1.9K evaluation images; (ii) multi-instances dataset which includes multiple instances with different visual characteristics within a single image, including 65K training images and 7K evaluation images. For both datasets, each image is accompanied by a corresponding language description.

*4.2.2 Evaluate data.* Apart from the validation set of the above two training datasets which totally contain 6.9k images, we use the first 5k images in ImageNet validation dataset, *i.e.* ImageNet-val5k. For the extended COCO-Stuff dataset and multi-instances dataset, each image is accompanied by several corresponding language descriptions. For ImageNet-val5k, we use BLIP [26] to generate a default text description of the color image to evaluate our method.

*4.2.3 Evaluation metric.* Following UniColor [16], we use Frechet Inception Distance (FID) [14] and colorfulness [13] to quantitatively evaluate the quality of our colorization results. We also utilize Δcolorfulness that computes the absolute difference of ground truth and colorization results to indicate realism in advance. Moreover, to

evaluate the controllability of text descriptions, we calculate CLIP similarity score [31] between the prompts and colorization results. As image colorization is an ill-posed problem which may have multiple reasonable solutions, we do not adopt Peak Signal-to-Noise Ratio (PSNR) [17], Structural Similarity Index Measure (SSIM) [39] or Learned Perceptual Image Patch Similarity (LPIPS) [48] for precisely match pairs of colorized result and ground truth.

## 4.3 Comparisons

We make comparisons with both automatic colorization methods and language-based coolorization methods. For automatic colorization methods, we compare our method with CIC [47], ColorFormer [19], DDColor [22] and CT2 [40]. For language-based colorization methods, we compare our methods with UniColor [16], L-CoDe [41], L-CoDer [5], L-CoIns [7] and L-CAD [6].

*4.3.1 Qualitative comparisons.* As shown in Fig. 6, we make comparisons with previous language-based colorization methods. Our

Table 1: Quantitative evaluation of different image colorization methods.

| Dataset | Extended COCO-Stuff | | | | Multi-instances | | | | ImageNet5k-val | | | |
|---|---|---|---|---|---|---|---|---|---|---|---|---|
| Metrics | FID↓ | Colorfulness↑ | ΔColorfulness↓ | CLIP Score↑ | FID↓ | Colorfulness↑ | ΔColorfulness↓ | CLIP Score↑ | FID↓ | Colorfulness↑ | ΔColorfulness↓ | CLIP Score↑ |
| CIC [47] | 38.41 | 36.25 | 7.20 | - | 30.77 | 33.78 | 7.10 | - | 20.45 | 26.34 | 12.11 | - |
| ColorFormer [19] | 16.84 | 42.25 | 1.20 | - | 6.24 | 37.17 | 3.71 | - | 6.16 | 37.63 | 0.82 | - |
| DDColor [22] | 7.49 | 40.97 | 2.48 | - | 14.21 | 44.54 | 3.66 | - | 6.77 | 41.55 | 3.10 | - |
| CT2 [40] | 18.81 | 44.21 | 0.76 | - | 10.64 | 41.21 | **0.34** | - | 7.50 | 39.68 | 1.23 | - |
| UniColor [16] | 10.51 | 40.92 | 2.69 | 25.14 | 9.00 | 34.17 | 6.71 | 24.75 | 10.63 | 34.98 | 3.47 | 23.96 |
| L-CoDe [41] | 31.82 | 29.21 | 14.24 | 29.93 | 27.89 | 25.22 | 15.66 | 27.59 | 19.56 | 24.23 | 14.22 | 29.10 |
| L-CoDer [5] | 31.03 | 29.68 | 13.77 | 29.91 | 24.04 | 24.79 | 16.09 | 27.50 | 16.73 | 22.90 | 15.55 | 29.39 |
| L-CoIns [7] | 37.34 | 35.71 | 7.74 | 29.64 | 27.89 | 33.53 | 7.35 | 27.61 | 21.03 | 33.53 | 4.92 | 29.52 |
| L-CAD [6] | 8.57 | 41.97 | 1.48 | 30.60 | 7.85 | 38.62 | 2.26 | 31.02 | 6.03 | 36.34 | 2.11 | 30.24 |
| ours-fantastic | 11.97 | **54.71** | 11.26 | 30.78 | 9.77 | **50.46** | 9.58 | 31.56 | 7.86 | **49.01** | 10.57 | 30.25 |
| ours-realism | **6.80** | 42.89 | **0.56** | **31.27** | **4.98** | 39.84 | 1.03 | **32.01** | **5.84** | 39.01 | **0.57** | **31.97** |
| ours-vintage | 9.65 | 38.79 | 4.66 | 30.13 | 5.27 | 35.40 | 5.48 | 30.24 | 9.62 | 37.53 | 0.92 | 29.97 |

method reduces color overflow better and generates more bright and colorful results. While L-CoDe [41], L-CoDer [5], L-CoIns [7] may suffer from color overflow of yellow sign on the forth row, or struggle with color under-saturation, UniColor [16] produces unreasonable colors (*e.g.* the green meat in the third row) when conditioned by complex text descriptions and inaccurate colors (*e.g.* brown ties in the second row should be red in the text description). L-CAD [6] can generate globally natural images, but it fails to reduce color overflow in some local area, such as the red tennis racket in the first row, the red color of tie overflows to the white shirt in the second row, and the unreasonable magenta knife in the third row. Additionally, although L-CAD utilizes diffusion model to generate high-realistic results, it still suffers from grayish and unnatural results, which can be seen obviously in the last row. The green leaf and trees behind two cats are recognized by our model and colorized to proper colors, while other methods can only generate gray or inproper colors.

*4.3.2 Quantitative comparisons.* As presented in Table 1, we present three different variants with different scaling factors to generate different styles of colors, ranging from fantastic ($\alpha = 1$), realism ($\alpha = 0.9$) to vintage ($\alpha = 0.8$). We make comparison with four unconditional method, CIC [47], ColorFormer [19], DDColor [22] and CT2 [40] to demonstrate our superior visual effect. The comparison between other five language-based colorization methods: Unicolor [16], L-CoDe [41], L-CoDer [5], L-CoIns [7] and L-CAD [6]. The results indicate our method can produce more plausible and accurate results. It is noticable that our method can not only generate high-realistic colorization results with the lowest FID and Δcolorfulness, but also generate the most colorful results with the highest colorfulness, satisfy the appetites of different users.

## 4.4 User Study

We further conduct user studies to evaluate the subjective perception of human observers. We invite 46 volunteers to answer 10 questions, each questions contains a text description and eight colorized results of previous methods and our results. We encourage participants to evaluate those colorized results from the following three aspects: (1) consistency with text descriptions; (2) realism of images; (3) personal preference. We present our results of different colorfulness ranging from fantistic, realistic, to vintage at the same time to evaluate the robustness of our COCO-Decoder. The

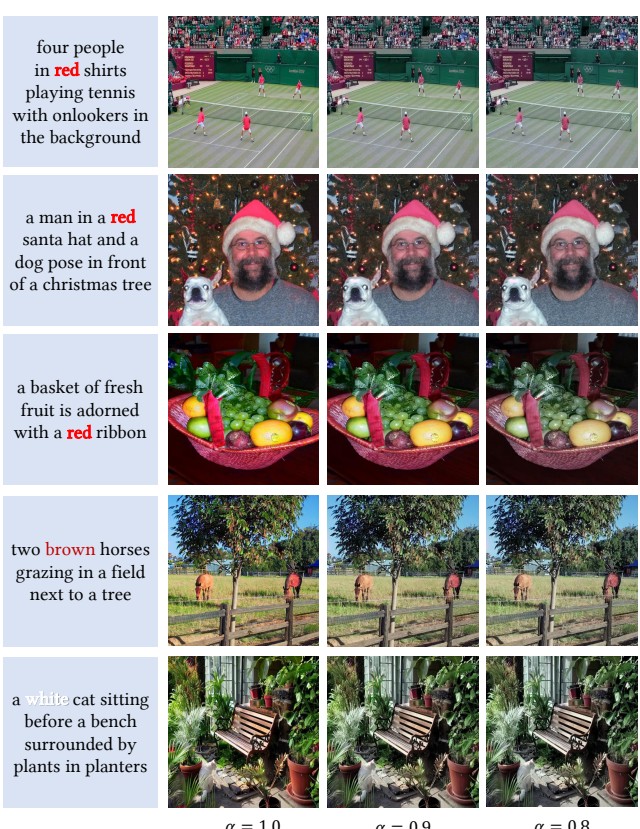

four people in **red** shirts playing tennis with onlookers in the background

a man in a **red** santa hat and a dog pose in front of a christmas tree

a basket of fresh fruit is adorned with a **red** ribbon

two **brown** horses grazing in a field next to a tree

a **white** cat sitting before a bench surrounded by plants in planters

$\alpha = 1.0$     $\alpha = 0.9$     $\alpha = 0.8$

Figure 7: Qualitative results of our Colorfulness Controllable Decoder (COCO-Decoder) with scaling factor $\alpha \in \{0.8, 0.9, 1.0\}$. Our results enjoy different color styles ranging from fantastic ($\alpha = 1.0$), realistic ($\alpha = 0.9$) to vintage ($\alpha = 0.8$). We allow users to control colorfulness in a simple and flexible way.

statistics results are presented as Fig. 8, which shows our method is prefered by most users.

## 4.5 Ablation Study and Discussion

We conduct other four baselines to demonstrate effectiveness of our coarse-to-fine framework and multi-level condition injection. The colorization results can be seen in Fig. 9.

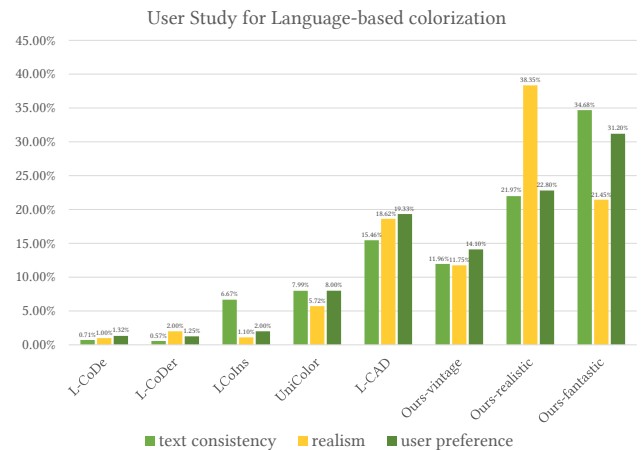

**Figure 8: Quantitative results of user study.**

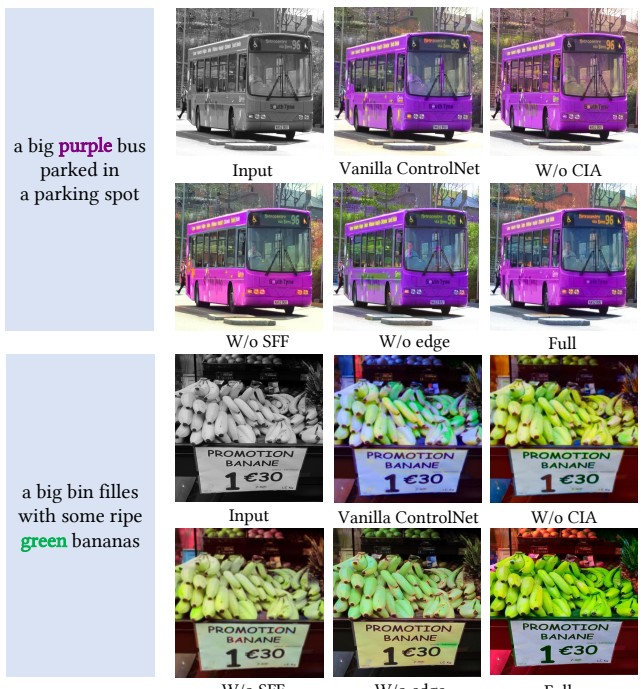

**Figure 9: Qualitative colorization results of ablation experiments. Zoom in for better visualization.**

**vanilla ControlNet**. We train a vanilla ControlNet without any design. We find experimentally that vanilla ControlNet will lead to color incomplete and color overflow. As shown in Fig. 9, tiny yellow spots appear on the bus body, and the number on the top of the bus is not completely colorized. The cloth of bus driver becomes purple due to the color overflow.

**Colorful Information Adaptor**. We remove CIA which insert CLIP color priors to latent codes of gray images. Without CIA, it's more difficult to build the accurate correspondence between color words and gray instances, so that the results become grayish or out of the control of text descriptions. In Fig. 9, the purple of the bus overflows to the light and the front glass, and the banana fails to create a link with "green" constaint.

**Semantic Feature Fusion**. We disable the insertion of the spatial semantic feature extract by Mask2Former [8] and CLIP [31]. As a result, obvious color overflow occurs in the colorized images. In Fig. 9, the bush behind the bus becomes purple and the paper sign under the bananas become green unexpectly.

**Low-level edge condition**. We disable the low-level edge condition, leading to color overflow and color incompletion.

**COCO-Decoder**. We present some results of different color styles by our colorfulness controllable decoder with different scaling factor, as shown at Fig. 7. Our method provides a user-friendly way of controllable color richness to generate diverse colors ranging from bland to gorgeous. Please refer to our supplementary materials for more visualization results.

In summary, we propose a coarse-to-fine framework that use CIA to insert rich color priors conditioned by language prompts in the coarse stage. On the basis of this semi-colorized result, we use high-level spatial semantic features and low-level edge latent codes to constrain colors to the correct area spatially. Disabling any one of them will lead to obvious color overflow.

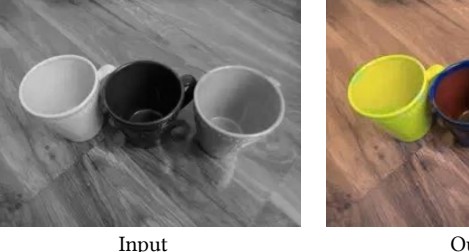

**Figure 10: Failure case of our method when dealing with multiple color words in a text prompt. Prompt: "the blue cup, the yellow cup and the cyan cup". The middle cup is incorrectly rendered red.**

## 5 CONCLUSION AND DISCUSSION

In this paper, we present **COCO-LC**, a novel coarse-to-fine framework that achieves COlorfulness COntrollable Language-based Colorization. We design a multi-level condition to reduce color overflow and COCO-Decoder to generate colorized results with diverse color styles flexibly. Extensive experiments demonstrate the superiority of COCO-LC over state-of-the-art image colorization methods in accurate, realistic and controllable colorization.

**Limitations and Future Work.** Stable Diffusion utilizes CLIP to align text and image domain, which struggles with complex text descriptions. When there are multiple colors and instances, it is hard to find accurate correspondence, leading to color-instance mismatch, as shown in Fig. 10. The color of the middle blue cup turns red unexpectly. In our future work, we would like to adopt more powerful cross-modality models and generative backbones to enhance the robustness of colorization.

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
