# OpenReview forum: "COCO-LC: Colorfulness Controllable Language-based Colorization"
_acmmm.org/ACMMM/2024/Conference — MM2024 Poster_

### Official Review · Reviewer_puSC · 2024-05-22

**Rating:** 5
**Confidence:** 3

**Summary:**

This paper introduces COCO-LC, a Language-based Colorization framework, which strengthens the image-text correlation by providing coarsely colorized results. The framework utilizes both low-level and high-level cues from the grayscale image to achieve precise, semantic-aware colorization without color overflow.

**Strengths:**

1. The colorization results closely align with the semantic colors as described in the language.
2. The results look more realistic than other methods in Figure 6.
3. The experiments is convincing to me.

**Limitations:**

1. It is unclear that the CLIP features of the color from the language can constrain the generation during the training process.
2. The two horses in Figure 7 are depicted with different colors, reflecting the description in the language as “two brown horses grazing in a field next to a tree.”
3. It lacks comparison of computational complexity.

**Suitability:**

3

---

### Official Review · Reviewer_67Gg · 2024-05-24

**Rating:** 4
**Confidence:** 3

**Summary:**

This paper focuses on Language-based image colorization. A method, named COCO-LC, consists of coarse-to-fine colorization, multi-level condition and colorfulness controllable colorization is proposed. Quantitative evaluation, user study and visual samples are provided

**Strengths:**

- The proposed method is well motivated and reasonable, the paper is well-written and easy to follow.
- Quantitative evaluation, user study and visual samples are provided.
- The proposed method achieved good performance.

**Limitations:**

- The colorization requires several steps and may result in significant time costs, which is not discussed in the text.
- Since color guidance is given only through text prompts and a pre-trained CLIP is used, the effectiveness of the method seems to depend heavily on the ability of the CLIP to map colors, which has not been specifically fine-tuned. This may lead to unstable final generation results. In addition, text itself is a very coarse representation of color control, which may result in the method not applicable to fine control tasks.
- A typo in line 398: "adptor" should be "adaptor"

**Suitability:**

3

---

### Official Review · Reviewer_CSkw · 2024-05-25

**Rating:** 2
**Confidence:** 3

**Summary:**

The paper tackles language-based image colorization, which converts grayscale images into visually appealing color images with language guidance. The authors introduce a new framework , which improves image-text correspondence using a coarse-to-fine approach. It employs a multi-level condition leveraging both low-level and high-level cues from the grayscale image for accurate colorization without overflow. It also introduces a scale factor to control the colorfulness of the output.

**Strengths:**

The results presented in the paper demonstrate good visual performance.
The quantitative metrics demonstrate that the proposed method surpasses previous approaches.

**Limitations:**

1. The role of the CIA module is technically concerning. As illustrated in Figures 2(a) and 3, the colors predicted by the CIA module are inaccurate. The original image depicts a purple bus, whereas the CIA predicts a red bus. How can features with such inaccurate representation enhance the alignment between color terms and the coloring results?
2. The intervals for the scaling factors shown in the Colorfulness Controllable Decoder (COCO-Decoder) are too small to observe any significant differences. Moreover, this method is very similar to the analysis of color properties in the latent space mentioned in the paper "Diffusing Colors: Image Colorization with Text Guided Diffusion", lacking innovation. Additionally, Figure 2 demonstrates that the COCO-Decoder does not require training, yet in Section 3.5, the authors mention a trainable decoder. This makes the pipeline of this module difficult to understand.
3. The innovation of the Consistency-aware Multi-level Condition module is also limited. The inclusion of edge information and semantic segmentation information has been widely used in previous coloring methods, and this paper does not propose insightful improvements.
4. The fairness of the comparisons with other methods is concerning. In Figure 6, the results labeled "Ours" have noticeably higher resolution than the results of other methods. Moreover, the L-CAD mentions that all comparisons were conducted at a resolution of 256, while this paper states that the test images were scaled to a resolution of 512. Is this the reason why the proposed method outperforms the compared methods?

**Suitability:**

2

---

### Official Review · Reviewer_mMt1 · 2024-05-27

**Rating:** 3
**Confidence:** 3

**Summary:**

This paper introduces a COCO-LC model, which completes the text-controlled colorization process through a two-stage Coarse to Fine Colorization approach. Initially, it leverages the advantages of CLIP to train a Colorful Information Adaptor (CIA) for Coarse Colorization. Subsequently, it uses ControlNet for fine Colorization. Additionally, it employs Canny edge maps and semantic segmentation maps extracted from grayscale images to achieve Multi-level Condition constraints in colorization, thereby mitigating color overflow. Finally, a Decoder is designed to control color intensity.

**Strengths:**

1. The proposed coarse to fine colorization framework addresses the image colorization problem by gradually narrowing the gap between the grayscale image domain and the color image domain through two stages.
2. It maximally utilizes the feature information contained in the grayscale images (such as Canny edge maps and semantic segmentation maps) for colorization.
3. In the qualitative comparison results, the model demonstrates significant advantages in detailed colorization and text matching.

**Limitations:**

1. This paper divides the colorization stage into Coarse Colorization and Fine Colorization phases. A major question arises: if there's a conflict in the colorization results between these two phases, which phase does the model's result align with? For instance, if the Coarse Colorization phase doesn't produce correct colorization results, does the subsequent Fine Colorization phase enhance the erroneous results, or does it primarily aim to correct them?

2. In line 165, it's mentioned that the model in the paper can ensure the process of "from vintage to gorgeous styles" by adjusting the Scaling factor 𝛼. Is this process merely adjusting the intensity of color features, as shown in Figure 7, or does it directly change the colorization style?

3. In line 438, it's stated that during the training of CIA, features  are obtained from the CLIP image encoder as f_I=E_I^CLIP (I). However, during testing, features E_T^CLIP (I) obtained from the CLIP image encoder are used instead of f_I. Could this direct replacement lead to uneven feature distribution, considering that the text here mainly focuses on color-related text rather than the entire image?

4. There's a lack of contrast images showing the same grayscale image guided by different text instructions for different colors (only one case is presented on the first page, which is insufficient).

5. Evaluation metrics are missing. Although in section 4.2.3, it's mentioned that metrics like PSNR, SSIM, and LPIPS are omitted due to multiple possible solutions in image colorization, comparing these metrics for image quality and factuality after colorization is necessary, similar to the comparison results in L-CAD[1].

6. In the ablation experiments, it's not clear whether the colorization effect follows the Coarse to Fine stages; therefore, only results using CIA should be presented to clarify this.

[1]	Weng S, Zhang P, Li Y, et al. L-CAD: Language-based Colorization with Any-level Descriptions using Diffusion Priors[J]. Advances in Neural Information Processing Systems, 2024, 36.

**Suitability:**

3

---

### Meta-Review · Area_Chair_Cehy · 2024-07-04

**Recommendation:** Accept (Poster)
**Confidence:** 5

**Metareview:**

The reviewers acknowledged the good results. The initial scores are all mixed with Borderline Reject, Weak Reject, Borderline Accept, Weak Accept. The main concerns from all reviewers are regarding the effectiveness of the CIA module and the lack of ablation and evaluation metrics. The authors tried to respond to these concerns, but not all reviewers were satisfied with the rebuttal. All the scores remained the same after the rebuttal. After careful discussion and consideration, the ACs decided to accept this paper for its strength in the results in the colorization task.

The authors are required to consider all the concerns raised by the reviewers, include the information added in the rebuttal phase (e.g., PSNR / SSIM / LPIPS metrics, time cost, ablation of CIA, etc.), and add necessary studies to further analyze the modules. Regarding PSNR/SSIM, the method is less favored. The authors are suggested to provide analysis if they focus too much on colorfulness rather than faithfulness. Reviewer CSkw left one concern with the color bleeding issue. The AC has checked the results and confirmed it happens but not seriously enough. Color bleeding is a common issue in colorization; it might stem from segmentation or edge issues. The authors are encouraged to discuss and analyze how color bleeding is reliant on segmentation and edge as well.